# Pressure Oxidation of Arsenic (III) Ions in the $H_3AsO_3$-$Fe^{2+}$-$Cu^{2+}$-$H_2SO_4$ System

**Kirill Karimov \*** , **Denis Rogozhnikov** , **Oleg Dizer, Maksim Tretiak** , **Sergey Mamyachenkov** and **Stanislav Naboichenko**

Department of Non-Ferrous Metals Metallurgy, Ural Federal University, 620002 Yekaterinburg, Russia; darogozhnikov@yandex.ru (D.R.); oleg.dizer@yandex.ru (O.D.); mtretyak144@gmail.com (M.T.); s.v.mamiachenkov@urfu.ru (S.M.); s.s.naboichenko@urfu.ru (S.N.)
\* Correspondence: kirill_karimov07@mail.ru; Tel.: +7-912-695-3175

**Abstract:** The processing of low-grade polymetallic materials, such as copper–zinc, copper–lead–zinc, and poor arsenic-containing copper concentrates using hydrometallurgical methods is becoming increasingly important due to the depletion of rich and easily extracted mineral resources, as well as due to the need to reduce harmful emissions from metallurgy, especially given the high content of arsenic in ores. Ferric arsenates obtained through hydrothermal precipitation are the least soluble and most stable form of arsenic, which is essential for its disposal. This paper describes the investigation of the oxidation kinetics of As (III) ions to As (V) which is required for efficient purification of the resulting solutions and precipitation of low-solubility ferric arsenates. The effect of temperature (160–200 °C), the initial concentration of Fe (II) (3.6–89.5 mmol/dm$^3$), Cu (II) (6.3–62.9 of mmol/dm$^3$) and the oxygen pressure (0.2–0.5 MPa) on the oxidation efficiency of As (III) to As (V) was studied. As (III) oxidation in $H_3AsO$-$Fe^{2+}$-$Cu^{2+}$-$H_2SO_4$ and $H_3AsO$-$Fe^{2+}$-$H_2SO_4$ systems was controlled by a chemical reaction with the apparent activation energy (Ea ($\approx$84.3–86.3 kJ/mol)). The increase in the concentration of Fe (II) ions and addition of an external catalyst (Cu (II) ions) both have a positive effect on the process. When Cu (II) ions are introduced into the solution, their catalytic effect is confirmed by a decrease in the partial orders, Fe (II) ions concentration from 0.43 to 0.20, and the oxygen pressure from 0.95 to 0.69. The revealed catalytic effect is associated with a positive effect of Cu (II) ions on the oxidation of Fe (II) to Fe (III) ions, which further participate in As (III) oxidation. The semi-empirical equations describing the reaction rate under the studied conditions are written.

**Keywords:** arsenic; iron; copper; oxidation; kinetics; catalytic effect; semi-empirical equation

## 1. Introduction

Under the conditions of depleting mineral reserves, copper and other non-ferrous metal producers are urged to employ various primary and industrial low-grade polymetallic materials, such as copper–zinc, copper–lead–zinc, poor arsenic-containing copper concentrates, middlings, etc. The overall decrease in the quality of mineral raw materials, along with the need to use arsenic-containing ores in processing, results in large volumes of various semi-products comprising this highly toxic element [1,2].

The current technologies for processing copper concentrates are mainly pyrometallurgical, resulting in the formation of a large amount of waste gases and dusts. Recycling of metallurgical dusts consists in extracting valuable components in the form of individual products, as well as in obtaining low-toxic compounds from harmful impurities [3–10]. Arsenic is a harmful impurity contained in recovered non-ferrous metals, which determines the importance of its removal from technological processes [11].

A large quantity of arsenic-containing wastes is formed during the processing of refractory gold concentrates [12], where arsenic is present in the forms of arsenopyrite, jarosite, scorodite, tennantite, etc. During pressure oxidation leaching of these materials, ferric arsenates are produced ($FeAsO_4 \cdot 2H_2O$, $FeAsO_4 \cdot 0.68$–$0.77H_2O$, $Fe(AsO_4)_x(SO_4)_y(OH)_z \cdot wH_2O$,

where $0.36 \leq x \leq 0.69$, $0.19 \leq y \leq 0.5$, $0.55 \leq z \leq 0.8$ and $0.2 \leq w \leq 0.45$). Their crystal structure depends on both the process temperature and the concentration of iron and other ions in the solution [13].

Considering the high environmental hazard of arsenic [14–16], its conversion to low-soluble, low-toxic compounds is an important research task [17]. Ferric arsenates obtained through hydrothermal precipitation are the least soluble and most stable form of arsenic, which is essential for its disposal [13,18]. According to [19], at temperatures below 100 °C and at a pH close to neutral, an amorphous solid phase precipitates, comprising a gel-like ferric hydroxide with adsorbed arsenate ions. As temperature increases and pH decreases, the crystalline structure of the precipitate improves; adsorbed arsenate ions react with iron (III) to yield a crystalline iron arsenate structure. This interaction occurs at 150–200 °C and at a Fe:As ratio of 1.5 and higher. The hydrothermal interaction affords $FeAsO_4 \cdot 2H_2O$, $Fe_3(AsO_4)_2SO_4OH$, $FeSO_4OH$, and $Fe_2(HAsO_4)_3$ $xH_2O$. At temperatures of 150 °C and above, over 95% of arsenic is removed. Thus, the precipitate obtained in experiments followed at 190 °C [19] still contain crystalline scorodite.

During the hydrometallurgical processing of industrial arsenic-containing materials, a large proportion of arsenic dissolves into the solution as As (III). The oxidation of arsenic (III) ions to As (V) is required for efficient purification of the resulting solutions and precipitation of low-solubility ferric arsenates.

The hydrothermal precipitation of iron–arsenic cakes from model solutions containing iron (III) and arsenic (V) ions has been extensively discussed in the literature [20–27]; however, little information is available on the precipitation of arsenic (III) and its oxidation to As (V). Research into the oxidation kinetics of As (III) to As (V) ions under hydrothermal conditions can contribute to purification optimization of solutions obtained by processing various arsenic-containing materials.

## 2. Materials and Methods

### 2.1. Materials and Apparatus

All chemicals were of analytical grade. The As (III) and Fe (III) solutions were prepared by dissolving solid $As_2O_3$ and $Fe_2(SO_4)_3 \cdot 9H_2O$ in deionized water. $CuSO_4 \cdot 5H_2O$, $H_2SO_4$ and high-purity oxygen were also used from AO Vekton (S-Petersburg, Russia). Sulfuric acid was then added dropwise to adjust the pH to a certain value.

Experimental solutions were prepared under iron (II) and arsenic (III) concentrations of 4.8 and 3.7 of $g/dm^3$, respectively, to avoid the formation of insoluble ferric arsenates during their hydrothermal oxidation.

Laboratory experiments were carried out using 1 $dm^3$ autoclaves (Parr Instrument, Moline, IL, USA) equipped with electrical heating, mechanical stirring, and sampling systems.

The temperature of the solution during the experiments was maintained constant at a set value $\pm 1.0$ °C. The oxygen flow rate was controlled using two mass-flow controllers (Bronkhorst High-Tech BV Inc., Ruurlo, The Netherlands).

### 2.2. Analysis

The concentration of As (III) in solutions was determined by titration with potassium bromate ($KBrO_3$) and Fe (II) with potassium dichromate ($K_2Cr_2O_7$). Liquid samples containing both As (III) and Fe (II) ions were titrated after separation in ion exchange columns [28]. To determine the Fe (II) concentration, the sample solution was transferred into an Erlenmeyer flask. Using a graduated cylinder, 25 mL of 1 M $H_2SO_4$ was added to the flask. Then, 10 $cm^3$ of the phosphoric acid solution and 8 drops of sodium diphenylamine sulfonate indicator were added to the flask. The intense purple color produced by the first drop of excess $K_2Cr_2O_7$ signals the end point for titration.

Samples from each experiment were taken at predefined time intervals and analyzed using inductively coupled plasma mass spectrometry (ICP-MS—NexION 300S quadrupole mass spectrometer, PerkinElmer Inc., Waltham, MA, USA).

A Pourbaix diagram was constructed using the HSC Chemistry Software Version 6.0 (Outokumpu Research Oy, Finland).

Oxidation efficiency of arsenic (III) in solution was estimated by Formula (1):

$$\alpha_{As(III)} = \frac{\left(C_{As_{total}} - C_{As(III)}\right)}{C_{As_{total}}} \times 100 \tag{1}$$

### 2.3. Experimental Procedure

An autoclave was loaded with 0.6 dm$^3$ of a solution with a known composition and heated to the required temperature. Then, oxygen was introduced, and a mechanical agitator was turned on. Portions of the solution were collected at determined intervals in a sealed vessel, cooled to atmospheric temperature and analyzed. In all experiments, pH = 1.0 was maintained, since, according to [29], this pH value has no effect on the process.

After the completion of the experiments, the autoclave was quickly cooled down to 70 °C with cold water and then depressurized. The liquid specimens were sampled for further analysis.

## 3. Results and Discussion

### 3.1. Influence of Iron Ions on Arsenic (III) Oxidation under Hydrothermal Conditions

3.1.1. Influence of Fe (II) and Fe (III) Ions

A solution of 3.7 of g/dm$^3$ As (III) was oxidized under pressured oxygen for 40 min. Table 1 shows the results of As (III) oxidation by pressured oxygen without the addition of Fe (II) ions. The ranges for the temperature, oxygen pressure, and pH value in the experiments were 150–200 °C, 0.5–1.0 MPa and [H$_2$SO$_4$]$_0$ = 4–20 of g/dm$^3$, respectively. The results of preliminary studies showed that the hydrothermal oxidation of arsenic (III) ions by oxygen in acidic media was insignificant. These data show that, without iron ions in the solution, the As (III) oxidation was hindered, even under hydrothermal conditions at a temperature of 200 °C and oxygen pressure of 1 MPa.

**Table 1.** Oxidation efficiency of As (III) to As (V) in the absence of iron ions in the solution.

| No. | *t* (°C) | O$_2$ (MPa) | Cu (g/dm$^3$) | H$_2$SO$_4$ (g/dm$^3$) | Oxidation Efficiency of As (III) |
|---|---|---|---|---|---|
| 1 | 150 | 0.5 | 0 | 4 | 2.1 |
| 2 | 180 | 0.5 | 0 | 4 | 5.5 |
| 3 | 200 | 0.5 | 0 | 4 | 7.2 |
| 4 | 150 | 1 | 0 | 20 | 3.4 |
| 5 | 180 | 1 | 0 | 20 | 6.2 |
| 6 | 200 | 1 | 0 | 20 | 9.1 |
| 7 | 150 | 1 | 4 | 10 | 3.8 |
| 8 | 180 | 1 | 4 | 10 | 6.9 |
| 9 | 200 | 1 | 4 | 10 | 8.9 |

It is known that Cu (II) ions can catalyze oxidation reactions, for example, the Fe (II) to Fe (III) ions oxidation. The introduction of Cu (II) into the solutions had little effect on the oxidation of As (III) ions without Fe (II) and Fe (III) ions (Table 1).

The As (III) oxidation efficiency as a function of Fe (III) ion concentration at a temperature of 180 °C, duration of 40 min, pH = 1.0, As (III) = 3.7 g/dm$^3$ with no oxygen introduced into the system, is shown in Figure 1.

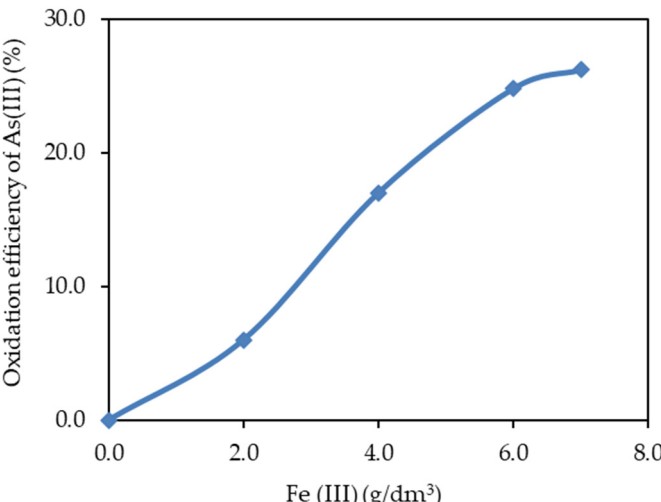

**Figure 1.** As (III) oxidation efficiency as a function of Fe (III) ion concentration with no oxygen introduced into the system.

Upon increasing the concentration of Fe (III) ions in the solution from 2 to 7 $g/dm^3$, the As (III) oxidation efficiency increased from 6.1 to 26.2%. Despite the lower standard oxidation potential of Fe (III)/Fe (II) (+0.77 V) than that of oxygen (+1.23 V, $O_2/H_2O$), the oxidation of arsenic by Fe (III) ions was more efficient.

Figure 2 demonstrates the effect of iron (II) ions on the degree of As (III) oxidation at 180 °C, duration of 40 min, pH = 1.0, oxygen pressure of 0.2 MPa.

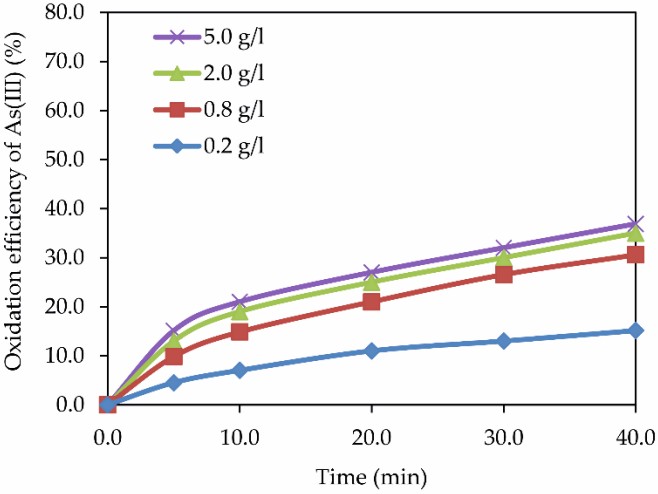

**Figure 2.** As (III) oxidation efficiency as a function of Fe (II) ion concentration.

Raising the initial concentration of iron (II) ions from 0.2 to 5.0 $g/dm^3$ had a positive effect on the As (III) ions oxidation efficiency, which increased from 15.3 to 36.8%.

Under hydrothermal conditions, Fe (II) ions are oxidized by oxygen to Fe (III), which further oxidizes As (III), according to the thermodynamic calculation presented in the following section.

### 3.1.2. Thermodynamics of As (III) Ions Oxidation in the Presence of Fe (III) Ions under Hydrothermal Conditions

The pH-dependences of the oxidation potentials of As (III) to As (V) and Fe (II) to Fe (III) couples are presented in a Pourbaix diagram (Figure 3).

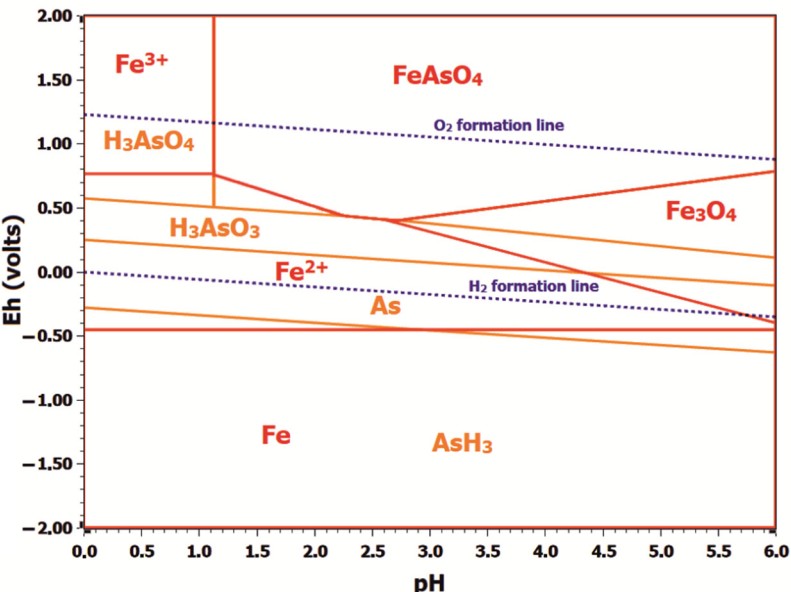

**Figure 3.** Pourbaix diagram of As-Fe-H$_2$O system.

According to the Pourbaix diagram (Figure 3), in acidic media (pH = 0–2), the complete oxidation of As (III) to As (V) occurred at Eh = 0.48–0.56 V in the As–H$_2$O system. At the same time, in the Fe–H$_2$O system, the oxidation of Fe (II) ions to Fe (III) occurred at Eh = 0.771 V. It should be noted that arsenic oxidation by Fe (III) ions was more efficient than that by oxygen (1.23 V, O$_2$/H$_2$O) [28].

The effect of iron (III) ions on arsenic oxidation may be associated with the formation of intermediate stable complexes of FeH$_2$AsO$_4{}^{2+}$ in acidic media.

The standard electrode oxidation potential of arsenic is Eh$_0$ = 0.56 V, the oxidation is described by the following reaction [30,31]:

$$H_3AsO_3 + H_2O = H_3AsO_4 + 2H^+ + 2e; \quad Eh_0 = 0.56 \text{ V} \tag{2}$$

The stability constant of the FeH$_2$AsO$_4{}^{2+}$ complex is equal to $4.36 \times 10^{-4}$, while the complex dissociation is described by the following reaction:

$$FeH_2AsO_4{}^{2+} + H^+ = Fe^{3+} + H_3AsO_4; \quad K_d = 4.36 \times 10^{-4} \tag{3}$$

At $t$ = 473 K, the oxidation potential of arsenic is calculated as follows:

$$Eh = 0.56 + 0.0195 \lg\left(\frac{a_{H_3AsO_4} \times a^2_{H^+}}{a_{H_3AsO_3}}\right) = 0.56 - 0.0390\text{pH} + 0.0204 \lg\left(\frac{a_{H_3AsO_4}}{a_{H_3AsO_3}}\right) \tag{4}$$

The stability constant is equal to:

$$K_d = \frac{a_{H_3AsO_4} \times a_{Fe^{3+}}}{a_{Fe_2H_2AsO_4^{2+}} \times a_{H^+}} = 4.36 \times 10^{-4} \tag{5}$$

By expressing activity $a_{H_3AsO_4}$ as $a_{H_3AsO_4} = \frac{K_d \times a_{H^+} \times a_{Fe_2H_2AsO_4^{2+}}}{a_{Fe^{3+}}}$ and placing it into an equation to calculate the oxidation potential of As (III), we obtain:

$$Eh = 0.56 - 0.0406\text{pH} + 0.0204 \lg\left(\frac{K_d \times a_{H^+} \times a_{Fe_2H_2AsO_4^{2+}}}{a_{Fe^{3+}} \times a_{H_3AsO_3}}\right) \tag{6}$$

$$Eh = 0.49 - 0.0611\text{pH} + 0.0204 \lg\left(\frac{a_{Fe_2H_2AsO_4^{2+}}}{a_{Fe^{3+}} \times a_{H_3AsO_3}}\right) \tag{7}$$

Specifically, at pH = 1.5, the oxidation potential of arsenic is equal to:

$$Eh = 0.40 + 0.0204 \lg \left( \frac{a_{Fe_2H_2AsO_4^{2+}}}{a_{Fe^{3+}} \times a_{H_3AsO_3}} \right) \tag{8}$$

The oxidation potential of Fe (II) to Fe (III) does not depend on pH (Figure 3) and is described by the following equation:

$$Fe^{2+} = Fe^{3+} + e^- \tag{9}$$

At $t$ = 453 K ($Eh_0$ = 0.771 V):

$$Eh = 0.771 + 0.0897 \lg \left( \frac{a_{Fe^{3+}}}{a_{Fe^{2+}}} \right) \tag{10}$$

The results of thermodynamic assessment explain the oxidation of As (III) by Fe (III) ions in acidic solutions, at $t$ = 180 °C, without introducing oxygen into the system. When added to the solution, Fe (II) ions are oxidized by oxygen to Fe (III) ions, which oxidize As (III).

### 3.2. Influence of Cu (II) Ions on Fe (II) and As (III) Oxidation in Hydrothermal Conditions

As shown earlier, copper (II) ions have little effect on arsenic (III) oxidation without iron ions in the solution. However, according to the previous studies [19,32] Cu (II) catalyzes Fe (II) to Fe (III) transition and Fe (III) is an oxidizing agent for As (III). Figure 4 demonstrates the effect of copper (II) ions on the degree of Fe (II) and As (III) oxidation at a temperature of 180 °C, duration of 40 min, pH = 1.0, As (III) = 3.7 g/dm$^3$ and oxygen pressure of 0.2 MPa.

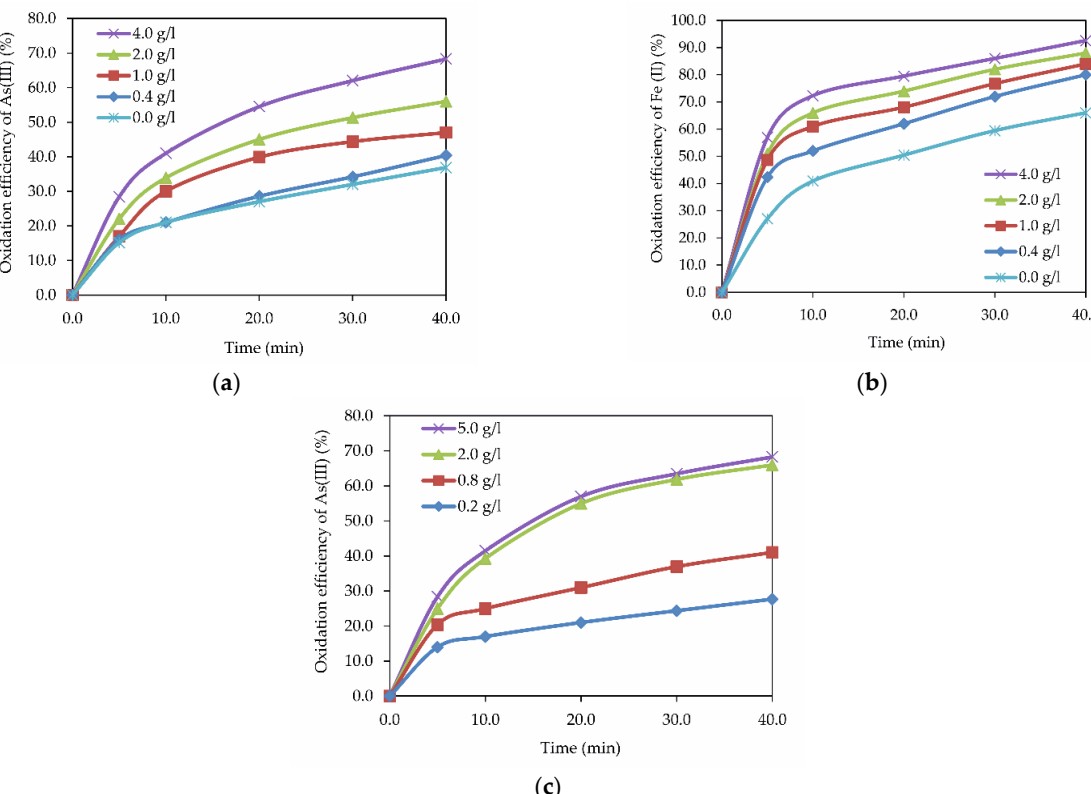

**Figure 4.** Oxidation efficiency of As (III) as a function of the copper concentration in the presence of iron (**a**), iron in the presence of copper (**b**), the oxidation efficiency of As (III) as a function of Fe (II) concentration in the presence of the copper (**c**).

According to the obtained data, addition of Cu (II) ions to the solution leads to a sharp increase in the As (III) oxidation efficiency in the presence of Fe (II) ions. If the concentration of Cu (II) in the solution increases from 0.4 to 4.0 g/dm$^3$, the oxidation efficiency of As (III) increases from 40.4 to 68.3%, and of Fe (II) from 66.2 to 92.6 % (Figure 4a,b).

An increase in the initial Fe (II) concentration from 0.2 to 5.0 g/dm$^3$ leads to an increase in the As (III) oxidation efficiency from 27.7 to 68.3%, relative to 15.3–35.1%, without adding copper to the solution. In the presence of Cu (II) ions, an increase in the iron concentration above 2 g/dm$^3$ has little effect.

An increase in the Cu (II) ion concentration from 0 to 4 g/dm$^3$ has a similar effect on the Fe (II) oxidation efficiency, which increases from 66 to 93%.

The positive effect of copper ions on As (III) oxidation is probably associated with an increase in the oxygen oxidizing power in the presence of variable valency ions (Co$^{3+}$, Fe$^{3+}$, Cu$^{2+}$), prone to complex formation or valency changes [19,28,32]. This explains the increase in the As (III) oxidation rate in the presence of Cu (II) ions. Cu (II) ions have a catalytic effect on Fe (II) to Fe (III) ion oxidation, necessary for the As (III) oxidation (reaction (11)). The catalytic effect of Cu (II) ions can be described by the following reactions:

$$Cu^{2+} + Fe^{2+} = Cu^+ + Fe^{3+} \qquad (11)$$

$$2Cu^+ + 1/2O_2 + 2H^+ = 2Cu^{2+} + H_2O \qquad (12)$$

Cu$^{2+}$ cations initiate the oxidation of Fe$^{2+}$ to Fe$^{3+}$ ions, which then oxidize As (III), as described by the following reactions:

$$H_3AsO_3 + 2Fe^{3+} + H_2O = 2Fe^{2+} + H_3AsO_4 + 2H^+ \qquad (13)$$

$$H_3AsO_4 + Fe^{3+} = FeH_2AsO_4^{2+} + H^+ \qquad (14)$$

### 3.3. Influence of Temperature on As (III) Ion Oxidation

Oxidation of As (III) in an oxygen atmosphere in the presence of Fe (II) ions with (4 g/dm$^3$) and without Cu (II) ions addition was performed across the temperature range of 160–200 °C, for 40 min, at pH = 1.0, As (III) = 3.7 g/dm$^3$, oxygen pressure 0.2 MPa and Fe (II) = 5.0 g/dm$^3$.

As shown in Figure 5, the As (III) oxidation efficiency at 160 °C slowly increased over time; however, it rapidly increased as the temperature increased to 200 °C, indicating a strong temperature effect on the As (III) to As (V) transition rate. Increasing the temperature from 160 to 200 °C leads to an increase in the As (III) oxidation efficiency from 18 to 45% and from 57 to 91%, without and with (4 g/dm$^3$) Cu (II) in the solution, respectively, in 40 min.

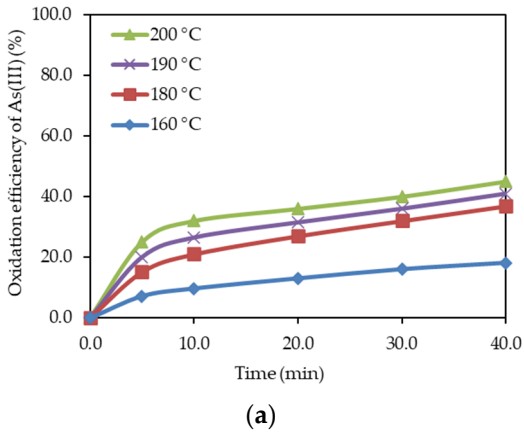 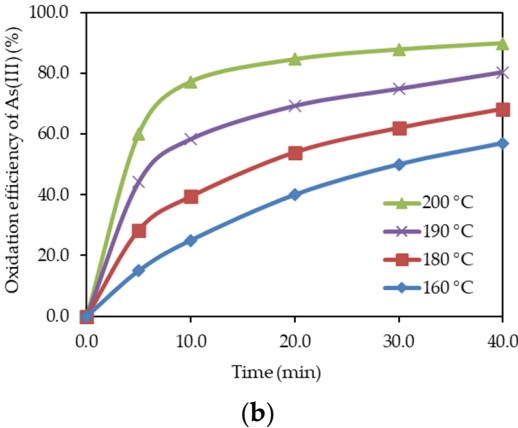

(a)      (b)

**Figure 5.** As (III) oxidation efficiency as a function of temperature with (**a**) and without (**b**) Cu (II) ions.

### 3.4. Influence of Oxygen Pressure on the Oxidation of Fe (II) and As (III) Ions

The influence of oxygen pressure on the As (III) and Fe (II) ion oxidation was assessed at a temperature of 180 °C, duration of 40 min, pH = 1.0, As (III) = 3.7 g/dm$^3$, oxygen pressure of 0.2–0.5 MPa and Fe (II) = 5.0 g/dm$^3$ (Figure 6).

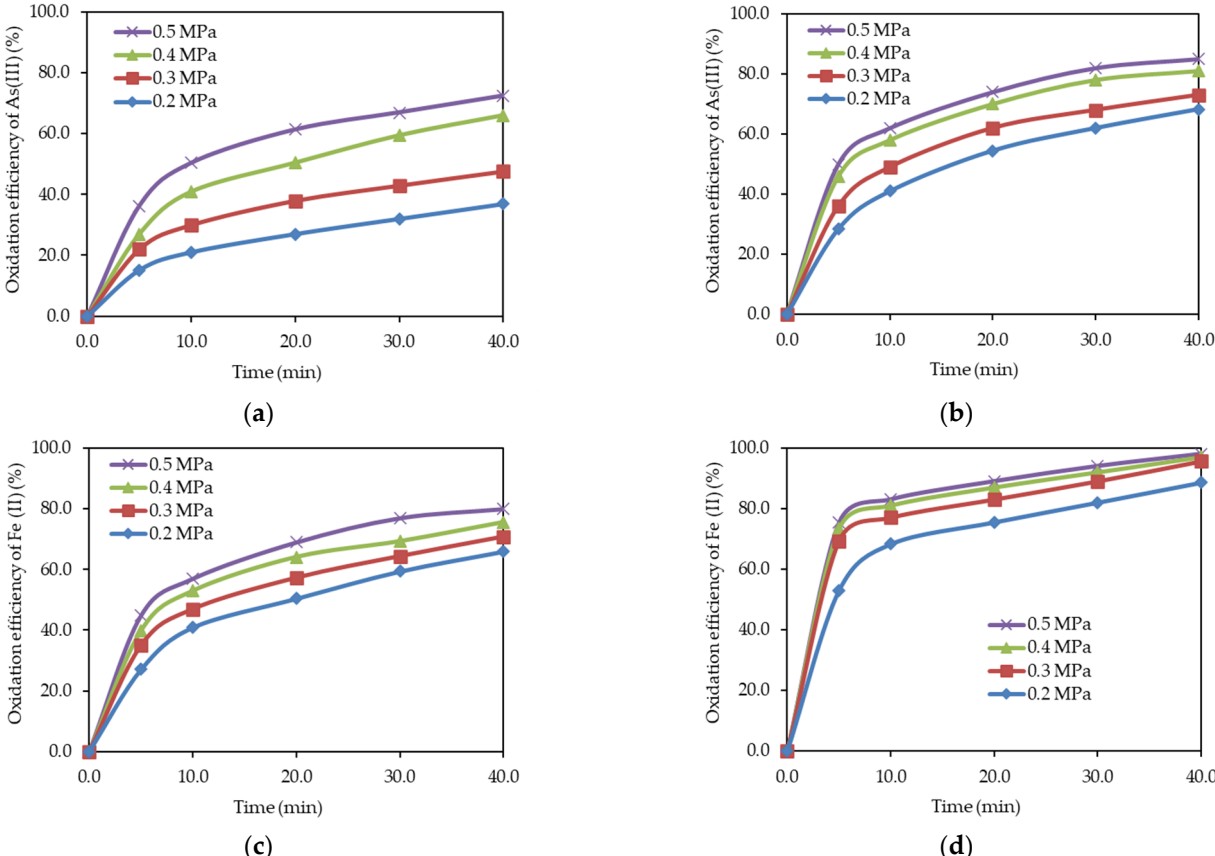

**Figure 6.** As (III) and Fe (II) oxidation efficiency as a function of temperature with (**a**,**c**) and without (**b**,**d**) Cu (II) ions.

Oxygen pressure has a significant effect on the As (III) and Fe (II) ions oxidation in solution. An increase in oxygen pressure from 0.2 to 0.5 MPa leads to an increase in the oxidation efficiency of As (III) and Fe (II), from 36.9 to 72.5% and 66.1 to 80.3%, respectively, within 40 min. Introduction of Cu (II) ions into the solution also had a positive effect on the oxidation efficiency. In addition, the influence of oxygen pressure on the process decreased; its increase from 0.2 to 0.5 MPa brought about an increase in the oxidation efficiency of As (III) from 68.3 to 85.2% and Fe (II) from 88.6 to 95.6%.

Increased oxygen pressure leads to a more intense oxidation of Fe (II) to Fe (III) ions, which further oxidizes As (III). Moreover, Cu (II) ions catalyze the Fe (II) to Fe (III) transition, hence being effective in reducing the oxygen pressure dependence of the process.

## 4. Kinetics Analysis

Since the As (III) oxidation efficiency is strongly influenced by temperature, Fe (II) concentration and oxygen pressure, this process can occur in both kinetic and diffusion regimes. This means that both the oxidizing agent diffusion into the reaction zone and the chemical reaction itself can be the rate-limiting stage of the process. An investigation of the process kinetics is necessary to determine the rate-limiting step.

A kinetics study of the oxidation of As (III) ions was performed at temperatures of 433–473 K (160–200 °C), duration of 40 min, pH = 1.0, As (III) = 13.3–100.6 mmol/dm$^3$, oxygen pressure of 0.2–0.5 MPa, Fe (II) = 3.6–89.5 mmol/dm$^3$, Cu (II) = 6.3–62.9 of mmol/dm$^3$.

The time-to-a-given-fraction method was used to calculate the kinetic parameters at different stages (As (III) oxidation efficiencies) [33]. The time required to achieve a certain degree of leaching and the apparent activation energy of $E_a$ are related according to Equation (15):

$$Lnt_x = const - lnA + E_a/RT \tag{15}$$

The slope angle of the graph displaying the relationship between $Lnt_x$ and $1/T$ allows the apparent activation energy to be calculated.

To determine the experimental activation energy of Cu-free As (III) oxidation, the inverse temperatures and natural logarithms of time were calculated (Figure 7a) at the following As (III) conversions in %: 5, 10, and 15. The same steps were implemented to calculate the experimental activation energy with copper ions (II) added to the solution (Figure 7b) at the following As (III) conversions in %: 15, 40, and 55. The experiment was repeated three times to obtain reliable values at each temperature; the averaged values are presented in Figure 7.

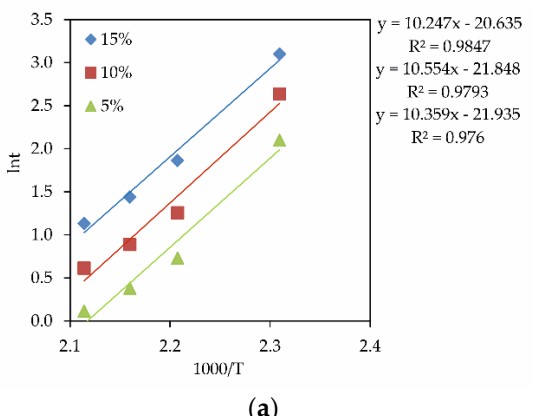
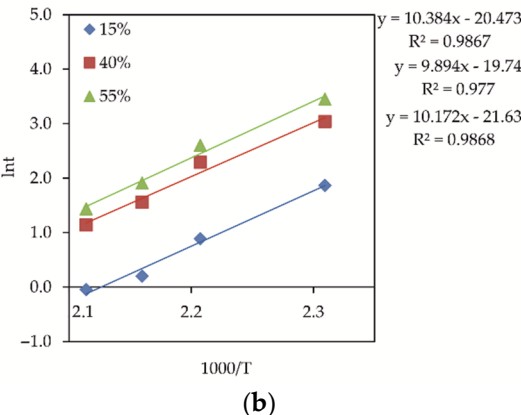

(a)             (b)

**Figure 7.** The correlation between $Lnt_x$ and $1/T$ for various As (III) oxidation efficiencies, without (**a**) and with (**b**) Cu (II).

The average values of the experimental activation energy of As (III) oxidation are 84.4 kJ/mol for the $H_3AsO_3$-$Fe^{2+}$-$Cu^{2+}$-$H_2SO_4$ system and 86.4 kJ/mol for the $H_3AsO_3$-$Fe^{2+}$-$H_2SO_4$ system, which are characteristic of systems controlled by a chemical reaction. In the course of the process, the values of the experimental activation energy undergo minor variations, indicating no changes in the process mechanism.

The differential method of initial rates was used to calculate partial orders. The initial As (III) oxidation rate (at $\tau = 0$) was determined from the plot of As (III) concentration (mmol/dm$^3$) against time (min). The initial rate was equal to the negative slope of the As (III) concentration curve against time at $\tau = 0$ min. In the study, the concentration of only one component was varied, while the others were added in excess and kept constant. To calculate partial orders, we plotted the relationship between the natural logarithms of the As (III) initial oxidation rates ($lnr_0$) and the natural logarithms of the concentrations of the studied components ($lnC_i$). The slope angle of the resulting plot corresponds to the partial order.

A graphical determination of the partial order relative to the concentration of iron ions is presented in Figure 8.

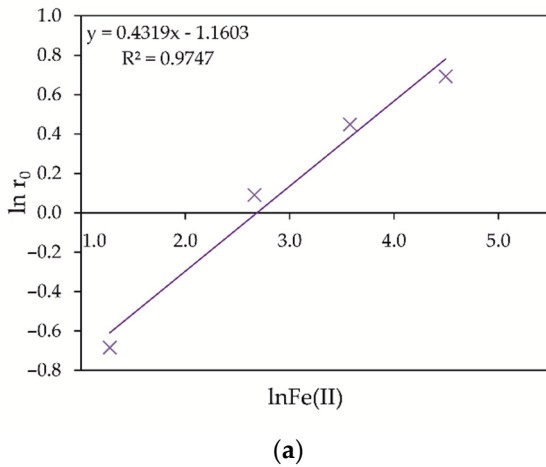

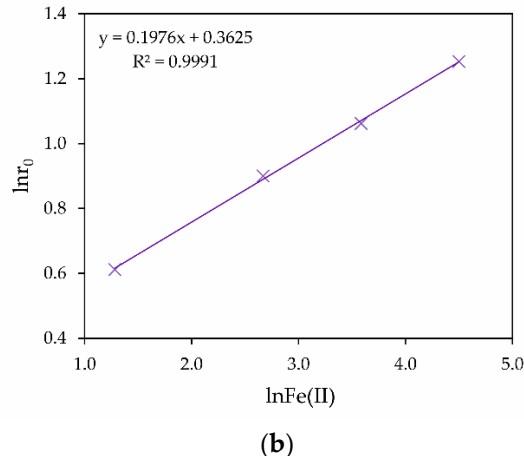

(**a**)

(**b**)

**Figure 8.** Dependence of $\ln r_0$ on $\ln Fe$ (II) without (**a**) and with (**b**) Cu (II).

The experimental reaction order relative to the concentration of Fe (II) ions is 0.43 for the $H_3AsO_3$-$Fe^{2+}$-$H_2SO_4$ system and 0.20 for the $H_3AsO_3$-$Fe^{2+}$-$Cu^{2+}$-$H_2SO_4$ system (Figure 8a,b, respectively). The difference in the reaction orders relative to iron was caused, apparently, by the catalytic influence of the Cu (II) ions on the Fe (II) ion oxidation in the $H_3AsO_3$-$Fe2^+$-$Cu^{2+}$-$H_2SO_4$ system (Equations (11)–(14)).

A graphical determination of the partial order relative to the Cu (II) ions concentration is given in Figure 9.

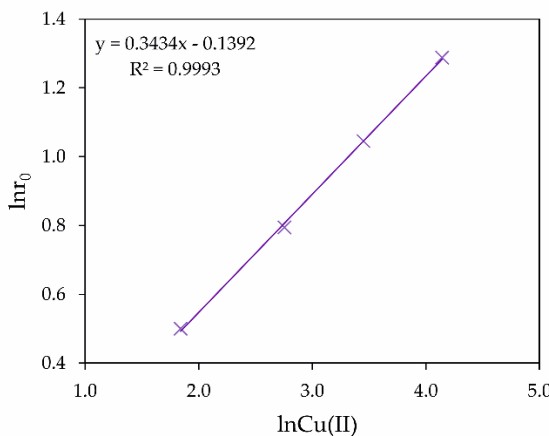

**Figure 9.** $\ln r_0$ versus $\ln Cu$ (II).

The experimental reaction order relative to the concentration of Cu (II) ions was 0.34 for the $H_3AsO_3$-$Fe^{2+}$-$Cu^{2+}$-$H_2SO_4$ system (Figure 9). The positive effect of copper ions on the As (III) oxidation was associated with the catalytic effect of oxygen on the Fe (II) ions oxidation, as mentioned earlier.

The reaction order relative to oxygen pressure was determined graphically (Figure 10).

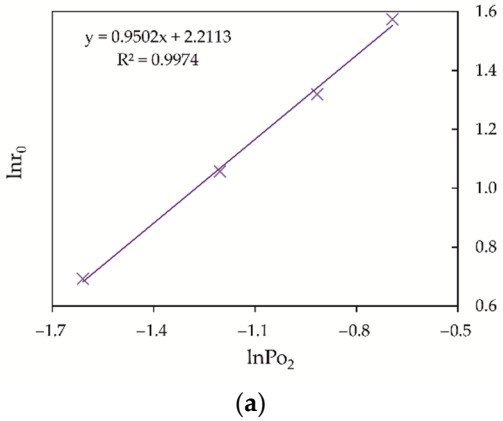
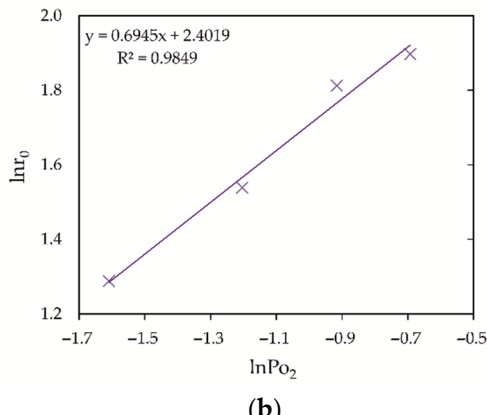

**(a)**                                    **(b)**

**Figure 10.** Dependence of $\ln r_0$ on $\ln Po_2$ without (**a**) and with (**b**) Cu (II).

According to the data obtained, the reaction order relative to oxygen pressure was 0.95 for the $H_3AsO_3$-$Fe^{2+}$-$H_2SO_4$ and 0.69 for the $H_3AsO_3$-$Fe^{2+}$-$Cu^{2+}$-$H_2SO_4$ system (Figure 10a,b, respectively). Introduction of Cu (II) ions into the solution led to a decrease in the partial order and, hence, to a decreased dependence of the process on oxygen pressure.

A graphical determination of the partial order relative to the concentration of As (III) ions is shown in Figure 11a,b. The experimental reaction order relative to the initial concentration of As (II) ions was 0.46 for the $H_3AsO_3$-$Fe^{2+}$-$Cu^{2+}$-$H_2SO_4$ system and 0.48 for the $H_3AsO_3$-$Fe^{2+}$-$H_2SO_4$ system.

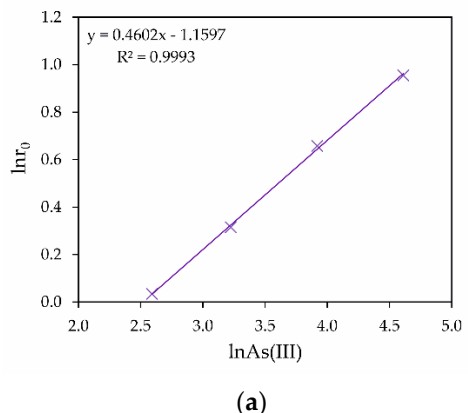
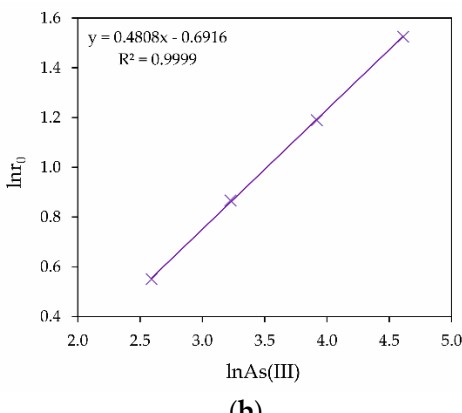

**(a)**                                    **(b)**

**Figure 11.** Dependences of $\ln r_0$ on $\ln As$ (III) without (**a**) and with (**b**) Cu (II).

To derive a general kinetic equation, we plotted all temperatures, concentrations, and oxygen pressures, which allowed the fixed slope angle of a = 1.08 to be determined. The "a" value obtained by the graphical method and the corresponding correlation coefficient $R^2$ are presented in Figure 12 for the $H_3AsO_3$-$Fe^2$-$H_2SO_4$ and $H_3AsO_3$-$Fe^{2+}$-$Cu^{2+}$-$H_2SO_4$ systems. The obtained value of the "a" coefficient corresponds to $k_0$. Some kinetic data at different parameters for oxidation efficiency of As (III) to As (V) are shown in Table 2.

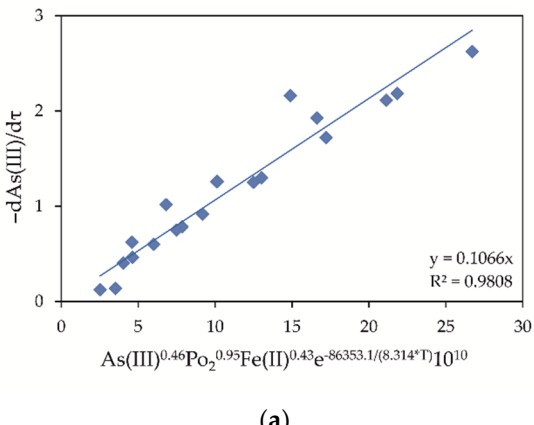

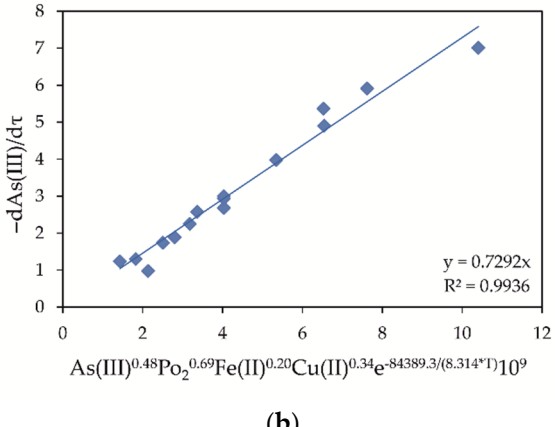

(**a**)                                                                                       (**b**)

**Figure 12.** Graphical determination of the k_o coefficient for $H_3AsO_3$-$Fe^{2+}$-$H_2SO_4$ (**a**) and $H_3AsO_3$-$Fe^{2+}$-$Cu^{2+}$-$H_2SO_4$ (**b**) systems.

**Table 2.** Kinetic data at different parameters for oxidation efficiency of As (III) to As (V).

| No. | $P_{O_2}$ (MPa) | Fe (II) (mmole/dm$^3$) | $t$ (K) | Cu (II) (mmole/dm$^3$) | $-dAs$ (III)/$d\tau$ |
|-----|------|-------|-----|-------|------|
| 1 | 0.2 | 90.01 | 433 | 0.00 | 0.14 |
| 2 | 0.2 | 90.01 | 463 | 0.00 | 1.92 |
| 3 | 0.2 | 3.58 | 453 | 0.00 | 0.12 |
| 4 | 0.2 | 35.81 | 453 | 0.00 | 1.02 |
| 5 | 0.3 | 90.01 | 453 | 0.00 | 2.16 |
| 6 | 0.4 | 90.01 | 453 | 0.00 | 2.23 |
| 7 | 0.2 | 90.01 | 453 | 6.29 | 1.30 |
| 8 | 0.2 | 90.01 | 453 | 31.47 | 2.25 |
| 9 | 0.2 | 90.01 | 433 | 62.95 | 1.24 |
| 10 | 0.2 | 90.01 | 463 | 62.95 | 4.91 |
| 11 | 0.2 | 3.58 | 453 | 62.95 | 0.98 |
| 12 | 0.2 | 35.81 | 453 | 62.95 | 2.58 |
| 13 | 0.3 | 90.01 | 453 | 62.95 | 3.98 |
| 14 | 0.4 | 90.01 | 453 | 62.95 | 5.37 |

Based on the above results, the following general kinetic equations can be derived for the hydrothermal oxidation of As (III) ions in the presence of Fe (II) ions:

For the $H_3AsO_3$-$Fe^{2+}$-$H_2SO_4$ system:

$$-\frac{dAs(III)}{d\tau} = 0.107 As(III)^{0.46} P_{O_2}^{0.95} Fe(II)^{0.43} e^{\frac{-86353.1}{8.314 \times T}} 10^{10} \tag{16}$$

For the $H_3AsO_3$-$Fe^{2+}$-$Cu^{2+}$-$H_2SO_4$ system:

$$-\frac{dAs(III)}{d\tau} = 0.729 As(III)^{0.48} P_{O_2}^{0.69} Cu(II)^{0.34} Fe(II)^{0.20} e^{\frac{-84389.3}{8.314 \times T}} 10^{9} \tag{17}$$

One can assume that As (III) oxidation in the $H_3AsO_3$-$Fe^{2+}$-$Cu^{2+}$-$H_2SO_4$ and $H_3AsO_3$-$Fe^{2+}$-$H_2SO_4$ systems was controlled by a chemical reaction with the apparent activation energy ($E_a \approx 84.3$–86.3 kJ/mol) for the following reasons: the increase in the concentration of Fe (II) ions and addition of an external catalyst (ions Cu (II)) had a positive effect on the process; elevated $E_a$ values in the reaction in the order of 0.43–0.20. When Cu (II) ions were introduced into the solution, their catalytic effect was confirmed by a decrease in partial orders, Fe (II) ions concentration from 0.43 to 0.20 and the oxygen pressure from 0.95 to 0.69. The revealed catalytic effect was associated with a positive effect of Cu (II) on the oxidation of Fe (II) to Fe (III) ions (Equations (11)–(14)), which further participate in the As (III) oxidation. This also confirms that Fe (III) ions formed that oxidized As (III).

## 5. Conclusions

This study focuses on the pressure hydrothermal oxidation of arsenic (III) ions in the $H_3AsO_3$-$Fe^{2+}$-$Cu^{2+}$-$H_2SO_4$ system formed during processing of arsenic-containing sulfide non-ferrous metals. The main conclusions of the study are as follows:

- Upon hydrothermal oxidation, Fe (II) ions are oxidized by oxygen to Fe (III) ions, which act as As (III) oxidizing agents. The influence of Fe (III) ions on arsenic oxidation can be associated with the formation of stable intermediate $FeH_2AsO_4^{2+}$ complexes in acidic media, which reduce the standard oxidation potential of As (III) to 0.40 V.
- It has been shown that the positive effect of copper ions on As (III) oxidation is associated with an increase in the oxygen oxidizing power in the presence of variable valency ions, prone to complex formation or valency changes. This explains the increase in the As (III) oxidation rate in the presence of Cu (II) ions. In addition, Cu (II) ions promote the oxidation of Fe (II) to Fe (III) ions.
- General kinetic equations for the As (III) oxidation rate in the $H_3AsO_3$-$Fe^{2+}$-$Cu^{2+}$-$H_2SO_4$ and $H_3AsO_3$-$Fe^{2+}$-$H_2SO_4$ systems were determined. The experimental reaction orders relative to arsenic and iron concentrations and oxygen pressure were obtained.
- As (III) oxidation in the $H_3AsO$-$Fe^{2+}$-$Cu^{2+}$-$H_2SO_4$ and $H_3AsO_3$-$Fe^{2+}$-$H_2SO_4$ systems was controlled by a chemical reaction with the apparent activation energy ($E_a$ ($\approx$84.3–86.3 kJ/mol)) for the following reasons: the increase in the concentration of Fe (II) ions and addition of an external catalyst (Cu (II) ions) both have a positive effect on the process; at a reaction order of 0.43–0.20, $E_a$ values are elevated. When Cu (II) ions are introduced into the solution, their catalytic effect is confirmed by a decrease in the specific orders of Fe (II) ions and the oxygen pressure. The revealed catalytic effect is associated with a positive effect of Cu (II) ions on the oxidation of Fe (II) to Fe (III) ions, which further participate in As (III) oxidation. This confirms that the formed Fe (III) ions act as oxidizing agents towards As (III).

**Author Contributions:** Conceptualization, D.R.; data curation, S.M. and S.N.; investigation, O.D. and M.T.; writing—review and editing, K.K. All authors have read and agreed to the published version of the manuscript.

**Funding:** This work was financially supported by the Russian Science Foundation Project No. 20-79-00321.

**Institutional Review Board Statement:** Not applicable.

**Informed Consent Statement:** Not applicable.

**Data Availability Statement:** Data are contained within the article.

**Conflicts of Interest:** The authors declare no conflict of interest.

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
