# Peer review of "Pressure Oxidation of Arsenic (III) Ions in the H3AsO3-Fe2+-Cu2+-H2SO4 System"

_metals, doi:10.3390/met11060975_

Round 1
Reviewer 1 Report
The authors investigate the oxidation kinetics of As (III) to As (V) ions to facilitate the hydrothermal precipitation of low solubility arseniates formed during the processing of ores and industrial arsenic-containing materials. This is important for obtaining the purification of the solutions developed during this processing.
Thus, the manuscript deals with a relevant issue. Some moderate corrections are needed. I suggest some changes:
Line 1-2. Title. I suggest a small change: instead of and hyphen between the two systems it is better to write “and”.
Line 56. Replace sediment by precipitate.
Line 60-61. This sentence must be modified, e.g. Thus, precipitates obtained in experiments followed at 190 ºC [19] still contain crystalline scorodite.
Line 72 and following. The section materials and methods must be rewritten. Firstly, in this section, the part of materials is absent. A first 2.1. subsection on materials is necessary. The analysis subsection shold be presented after you explain how did you obtained the solutions.
Line 110. In the results and discussion section first you should explain was the result of an action followed by a discussion. In this line you begin with the conclusion and followed by the result.
Table 1. the sentence of the caption needs an end point.
Line 147. The Pourbaix diagram need a citation. All the 3.1.2. subsection has only one reference.
The section of conclusions needs be slightly improved. this is a summary not conclusions, it contains too many specific data.
Author Response
Dear Reviewer,
Line 1-2. Title. I suggest a small change: instead of and hyphen between the two systems it is better to write “and”.
This is one system. «Pressure oxidation of arsenic (III) ions in the H3AsO3–Fe2+–Cu2+–H2SO4 system».
Line 56. Replace sediment by precipitate.
Its corrected.
Line 60-61. This sentence must be modified, e.g. Thus, precipitates obtained in experiments followed at 190 ºC [19] still contain crystalline scorodite.
Its corrected.
Line 72 and following. The section materials and methods must be rewritten. Firstly, in this section, the part of materials is absent. A first 2.1. subsection on materials is necessary. The analysis subsection shold be presented after you explain how did you obtained the solutions.
2.1. Materials and apparatus
All chemicals were of analytical grade. The As (III) and Fe (III) solutions were prepared by dissolving solid As2O3 and Fe2(SO4)3 ·9H2O in deionised water. CuSO4 ·5H2O, H2SO4 and high purity oxygen were also used. Sulfuric acid was then added dropwise to adjust their pH to a certain value.
Experimental solutions were prepared under the iron (II) and arsenic (III) concentrations of 4.8 and 3.7of g/dm3, respectively, to avoid the formation of insoluble ferric arsenates during their hydrothermal oxidation.
Laboratory experiments were carried out using 1 dm3 autoclaves (Parr Instrument, Moline, IL, USA) equipped with electrical heating, mechanical stirring and sampling systems.
The temperature of the solution during the experiments was maintained constant at a set value ± 1.0 °C. The oxygen flow rate was controlled by using two mass-flow controllers (Bronkhorst High-Tech).
Line 110. In the results and discussion section first you should explain was the result of an action followed by a discussion. In this line you begin with the conclusion and followed by the result.
A solution of 3.7 of g/dm3 As (III) was oxidized under pressured oxygen for 40 min. Table 1 shows the results of As(III) oxidation by pressured oxygen without the addition of Fe(II) ions. The changing ranges for the temperature, oxygen pressure, and pH value in the experiments were 150–200 °C, 0.5–1.0 MPa and ([H2SO4]0 = 4-20 of g/dm3, respectively. The results of preliminary studies showed that the hydrothermal oxidation of arsenic (III) ions by oxygen in acidic media is insignificant. This data shows that, without iron ions in the solution, the As (III) oxidation is hindered even under hydrothermal conditions at a temperature of 200 °C and oxygen pressure of 1 MPa.
Table 1. the sentence of the caption needs an end point.
Its corrected.
Line 147. The Pourbaix diagram need a citation. All the 3.1.2. subsection has only one reference.
The Pourbaix diagram was constructed using the HSC Chemistry Software Version 6.0 (Outokumpu Research Oy, Finland). References is added.
The section of conclusions needs be slightly improved. this is a summary not conclusions, it contains too many specific data.
Its corrected.
Thank You for your comments and suggestions to improve our manuscript.
Reviewer 2 Report
significantly better, which is why I requested a major revision.
Is Cu a catalytic effect or a reagent? What is the Catalytic cycle?
The date in Table I does not seem to be referenced. It seems that "preliminary studies showed that the hydrothermal oxidation of arsenic" is the only explanation. The authors need to provide the background or just state that Cu does not work without Fe.
"Oxidation efficiency" should be formally defined. I assume that the ICP measurement of the As species are used to determine it. However, an equation should be presented to help the reader.
Is there a reference for the Pourbaix diagram (Eh–pH diagram) shown? You should state that you’re using the Nernst equation for this section and that positive is thermodynamically variable. Sources for number should be reference. Using log and lg so confusing, is it log10 or ln? I had problem finding how 0.0449 was computed. It may just be easier to ref 30 and state its conclusion.
It would be nice to have a table to summarize the kinetic data, i.e. concentrations and rates. This will help support the fit (15 and 16). Are their error bars from the non-linear fit?
Do the orders mean anything?
Author Response
Dear Reviewer,
Is Cu a catalytic effect or a reagent? What is the Catalytic cycle?
Cu (II) ions use for catalytic effect on the Fe (II) to Fe (III) ions oxidation, necessary for the As (III) oxidation. The catalytic cycle of the copper ions effect on the Fe (II) to Fe (III) ions oxidation is described by summarizing reactions 10, 11.
The date in Table I does not seem to be referenced. It seems that "preliminary studies showed that the hydrothermal oxidation of arsenic" is the only explanation. The authors need to provide the background or just state that Cu does not work without Fe.
A solution of 3.7 of g/dm3 As (III) was oxidized under pressured oxygen for 40 min. Table 1 shows the results of As(III) oxidation by pressured oxygen without the addition of Fe(II) ions. The changing ranges for the temperature, oxygen pressure, and pH value in the experiments were 150–200 °C, 0.5–1.0 MPa and ([H2SO4]0 = 4-20 of g/dm3, respectively. The results of preliminary studies showed that the hydrothermal oxidation of arsenic (III) ions by oxygen in acidic media is insignificant. This data shows that, without iron ions in the solution, the As (III) oxidation is hindered even under hydrothermal conditions at a temperature of 200 °C and oxygen pressure of 1 MPa.
It is known that Cu (II) ions can catalyse oxidation reactions, for example, the Fe (II) to Fe (III) ions oxidation. The introduction of Cu (II) into solutions had little effect on the oxidation of As (III) ions without Fe (II) and Fe (III) ions (Table 1).
"Oxidation efficiency" should be formally defined. I assume that the ICP measurement of the As species are used to determine it. However, an equation should be presented to help the reader.
It is added.
Is there a reference for the Pourbaix diagram (Eh–pH diagram) shown?
The Pourbaix diagram was constructed using the HSC Chemistry Software Version 6.0 (Outokumpu Research Oy, Finland).
You should state that you’re using the Nernst equation for this section and that positive is thermodynamically variable. Sources for number should be reference.
Its added.
Using log and lg so confusing, is it log10 or ln?
We used log10, its corrected.
I had problem finding how 0.0449 was computed. It may just be easier to ref 30 and state its conclusion.
It error in calculations. its corrected.
It would be nice to have a table to summarize the kinetic data, i.e. concentrations and rates. This will help support the fit (15 and 16).
Its added in table 2.
Are their error bars from the non-linear fit?
Its linear because correlation coefficient R2 = 0.98 and 0.99.
Do the orders mean anything?
When Cu (II) ions are introduced into the solution, their catalytic effect is confirmed by a decrease in partial orders, Fe (II) ions concentration from 0.43 to 0.20 and the oxygen pressure from 0.95 to 0.69.
Thank You for your comments and suggestions to improve our manuscript.
Round 2
Reviewer 1 Report
The authors revised all the suggested modifications and now I consider that the manuscript is OK.
Reviewer 2 Report
The authors made the suggested improvements. I appreciate their effort to improve their paper.